# Mpox Virus in the Pharynx of Men Having Sex with Men: A Case Series

**DOI:** 10.3390/pathogens13010092

**Published:** 2024-01-20

**Authors:** Silvia Limonta, Giuseppe Lapadula, Luca Mezzadri, Laura Corsico, Francesca Rovida, Alice Ranzani, Fausto Baldanti, Paolo Bonfanti

**Affiliations:** 1IRCCS “San Gerardo dei Tintori”, 20900 Monza, Italy; silvia.limonta@irccs-sangerardo.it (S.L.); laura.corsico@irccs-sangerardo.it (L.C.); paolo.bonfanti@unimib.it (P.B.); 2School of Medicine, University of Milano Bicocca, 20900 Milan, Italy; l.mezzadri3@campus.unimib.it; 3Department of Clinical, Surgical, Diagnostic and Paediatric Sciences, University of Pavia, 27100 Pavia, Italy; francesca.rovida@unipv.it (F.R.); fausto.baldanti@unipv.it (F.B.); 4SC Microbiology and Virology, IRCCS Policlinico San Matteo, 27100 Pavia, Italy

**Keywords:** Mpox, MsM, viral shedding, monkeypox

## Abstract

The recent Mpox virus (MPV) outbreak in Europe and North America, primarily among men who have sex with men (MSM), raised concerns about various transmission sources. We examined patients with Mpox from an urban STI center in Lombardy, Italy, between May and August 2022. Demographic, transmission, and clinical data were collected using a standardized form. Initial and subsequent tests were conducted using the RealStar Orthopoxvirus PCR Kit 1.0 (Altona Diagnostics, Hamburg, Germany) for skin lesions and oropharyngeal swabs. A total of 15 patients were recruited, all MSM, with 40% being HIV-positive. Almost all reported recent unprotected sexual activity. Oropharyngeal symptoms were observed in a minority, and oral cavity lesions were present in 20% of cases. MPV DNA was detected in skin lesions of 93% of patients and in oropharyngeal swabs of 87%. Skin samples exhibited a higher viral load than pharyngeal samples, with the latter persisting longer. Prospective follow-up of 11 individuals revealed an average pharyngeal persistence of 5.3 days beyond skin lesion clearance, reaching up to 80 days in an immunosuppressed case. Our findings indicate that MPV replication can persist in the pharynx asymptomatically and for an extended period.

## 1. Introduction

Since May 2022, a new outbreak of Mpox (formerly known as monkeypox), caused by the zoonotic orthopox DNA virus Mpox, has spread worldwide to non-endemic countries. On 23 July 2022, the World Health Organization (WHO) declared it a public health emergency of international concern [1]. As of October 2023, a total of 26,229 cases have been identified across 45 countries in the European region. Among these, 963 cases were reported in Italy [2].

The transmission of the virus can occur through various routes, including large respiratory droplets, close contact, and contaminated surfaces (fomites) [3]. Since 2022, however, the majority of cases have been linked to close contact during sexual activity among men who have sex with men (MSM) [4,5].

Mpox virus (MPV) DNA has been detected in respiratory specimens, skin swabs, urine, blood, and other biological samples [3]. However, the dynamics of viral persistence in body fluids and its clinical relevance are not yet fully understood.

In this case series, we describe the epidemiology, clinical features, and, specifically, the prevalence and persistence of MPV DNA in skin and oropharyngeal swabs.

## 2. Materials and Methods

We describe a case series of 15 patients who were consecutively diagnosed with Mpox between May and August 2022 at an urban sexually transmitted diseases (STD) center in Northern Italy.

Data about demographic characteristics, mode of transmission, and clinical presentation were retrospectively collected from patients’ charts using a standardized form.

Skin and oropharyngeal swabs were collected from all patients at baseline and tested for the presence of MPV DNA using the RealStar Orthopoxvirus PCR Kit 1.0 (Altona diagnostic, Hamburg, Germany).

Starting in July, all patients who tested positive underwent weekly follow-up testing for skin lesions and oropharynx until negative results were obtained.

## 3. Results

A total of 15 patients were included, all of whom were men who have sex with men (MSM). The median age was 36 years (range 35–45). Only two of them had a history of previous smallpox vaccination. Six patients (40%) were HIV positive, all of whom had HIVRNA < 50 copies/mL, and five out of ssix had a CD4+ T-cell count > 500 cells/mm^3^ (median (range) 749/mm^3^ (129–1002)). Two more patients reported comorbidities associated with immunodeficiency (one diabetes mellitus and one previous bone marrow transplantation). All but one patient reported recent unprotected sexual activity, with two cases involving only oral intercourse. Four patients had concurrent sexually transmitted diseases: two syphilis, one chlamydial infection, and one gonorrhea.

A complete record of patients’ characteristics are shown in Table 1.

Upon onset, systemic symptoms were common. Fever was reported by 12 patients (80%), and newly onset lymphadenopathy was reported by 9 (60%). None of the patients had previous lymphadenopathy or other systemic symptoms due to their underlying conditions, particularly HIV infection. In many cases, fever preceded the appearance of the cutaneous rash. Oropharyngeal symptoms, such as pharyngodynia or odynophagia, were reported only by one-third of the patients, and lesions in the oral cavity were present only in 20% of cases. Conversely, a significant number of patients (66.7%) experienced clinical manifestations of proctitis, with symptoms like anorectal pain, tenesmus, diarrhea, and bleeding. These symptoms were not always associated with visible lesions in the perianal area.

All patients presented with skin and/or mucosal lesions, often in the genital and perianal regions. The majority of cases were mild and self-limiting, and no patient required hospitalizations or specific therapy.

The complete clinical features of the patients are depicted in Table 2.

MPV DNA was identified from skin lesions in 14 out of 15 individuals and from oropharyngeal swabs in 13/15 (86.7%). In one case, the diagnosis relied exclusively on the results from the oropharyngeal swab. Conversely, in two instances, the baseline oropharyngeal swab yielded negative results.

The cycle threshold (Ct), that is, the number of cycles required to amplify viral DNA to detectable levels, which provides an estimate of viral load, was available for all samples. In all available paired skin and oropharyngeal samples, lower Ct values, indicating a higher amount of viral DNA, were found in the skin than in the pharynx (mean Ct 18.1 [95%CI 14.8–21.5] vs. 24.2 [21–27.5]).

Longitudinal follow-up was undertaken only for patients enrolled after July 2022, encompassing 11 out of 15 patients included in the original cohort. The results of the swabs conducted on both throat and skin lesions, along with the corresponding Ct values, are presented in Table 3. Among them, MPV DNA positivity persisted, on average, 9 days longer in the pharynx than in the skin (mean (SD): 26.3 (19.5) days *versus* 17.3 (5.4)). Of note, in one immunosuppressed patient, due to their previous bone marrow transplantation (Patient 1 in Table 3), oropharyngeal swabs remained positive for 80 days (55 days beyond skin lesion resolution).

## 4. Discussion

We described a human Mpox case series including 15 MSM from Italy reported over a three-month period in 2022. All cases were possibly infected through sexual intercourse, in line with the epidemiologic features of the 2022 global outbreak [5].

Significantly, nearly 90% of patients exhibited positive MPV DNA detection in oropharyngeal swabs, irrespective of their clinical presentation or symptomatic manifestation of oropharyngeal involvement. The presence of MPV in diverse biological fluids, including saliva, rectal fluid, and semen, has been previously documented [3], bolstering the notion that intimate contact with bodily fluids during sexual intercourse serves as a prominent route of transmission. Anorectal and genital epithelia have been postulated as preferential points of entry due to their diminished keratinization and heightened prevalence of antigen-presenting cells, such as macrophages and dendritic cells [6]. The potential for oropharyngeal mucosa to serve as an additional gateway or source of transmission warrants further comprehensive investigation.

In line with other studies [6,7,8,9], the patients in our series most commonly presented with lesions in the ano-genital area, often preceded by local lymphadenopathy. By contrast, despite the high frequency of pharyngeal swab positivity, only a minority of them had signs or symptoms of pharyngeal involvement, consistent with previous reports [10].

The incremental value of systematic oropharyngeal swab testing in cases of suspected Mpox or in the aftermath of viral exposure warrants further investigation. In a prior study conducted by Coppens and colleagues, the sensitivity of oropharyngeal swabs among individuals with active skin lesions was 64% [11]. Within our series of cases, sensitivity appeared to be even higher, approaching 90%. Notably, in a singular case, the oropharyngeal swab proved to be the sole means of establishing a conclusive diagnosis. Therefore, oropharyngeal swabs may potentially assume an additional diagnostic role in Mpox cases, particularly for patients lacking skin lesions or in the pursuit of early diagnosis post-exposure [12].

Our observations also revealed that MPV DNA remained detectable through PCR testing in upper respiratory tract swabs for an average of 5 days longer than in skin lesions. In one case involving an immunosuppressed patient, viral DNA persisted for up to 80 days. Our findings are consistent with a study conducted in Spain, where viable virus shedding was reported from the oropharynx and PCR-positivity persisted for up to three weeks [13]. Similar to our results, the viral DNA loads were lower in the pharynx than in skin lesions in the cited study. We also found that viral DNA measured by Ct values was significantly lower in pharyngeal swabs than in skin and mucosal lesions. Unlike the mentioned study, however, in our observation, the decay of viral DNA in the oropharynx appeared to be slower than in the skin. Due to the susceptibility of Ct values to various factors, such as sample volume and cell numbers, comparing Ct values between different sample sources can be challenging. In addition, the knowledge of the dynamics of MPXV replication and the change of Ct values during the course of Mpox disease is still limited. Despite these limitations, Ct values may serve as an approximation of viral load and aid in understanding the viral dynamics across diverse biological samples [14].

The role of oropharyngeal secretions in viral transmission, particularly from patients with persistent viral replication in the pharynx beyond the healing of skin lesions, remains unclear. If the oropharynx secretions were indeed involved in viral transmission, one would expect larger outbreaks to occur beyond sexual contexts or instances of infections within households. However, none of the patients under investigation reported household infections or transmission occurring to anyone other than their sexual partners, indicating that this mode of transmission is unlikely. Therefore, further studies are warranted to investigate the potential role of oropharyngeal secretions in Mpox transmission.

Our study has some limitations. Firstly, the number of participants enrolled was small. Secondly, the study has a retrospective design, which may introduce inherent biases. Additionally, it is important to note that after the first appearance of negative results, we did not perform further swabs. This may have led to a potential missed detection of a relapse. Eventually, we did not perform a viral culture; therefore, we could not explore the association between the presence of viral DNA and the shedding of the viable infectious virus from the respiratory tract.

## 5. Conclusions

Our findings suggest that MPV DNA can be detected in the pharynx, even in individuals who do not display any symptoms or indications of local infection. The virus exhibits prolonged persistence in the pharyngeal region, especially among immunosuppressed patients. Regular screening of the oropharyngeal swab in suspected cases and those who have been exposed could potentially enhance the effectiveness of diagnosis, even in the absence of active skin or mucosal lesions. Whether replication in the oropharynx may contribute to infection transmission needs to be further assessed.

## Figures and Tables

**Table 1 pathogens-13-00092-t001:** Demographic and clinical characteristics of patients diagnosed with Mpox.

Characteristic	N (%) or Median (IQR)
Male gender	15 (100%)
Age (years)	36 (35–45)
Men having sex with men	15 (100%)
Comorbidities	8 (53.4%)
HIV positive	6 (40%)
Bone marrow transplantation	1 (6.7%)
Diabetes	1 (6.7%)
Vaccinated for smallpox	2 (13.3%)
Reporting recent unprotected sex	14 (93.3%)
Receptive anal intercourses	1 (6.7%)
Receptive anal and oral intercourses	2 (13.3%)
Only oral sex	2 (13.3%)
Sex with other men (not specified)	9 (60%)
Other concurrent sexually transmitted infections	
Syphilis	2 (13.3%)
Chlamydia	1 (6.7%)
Neisseria gonorrhoeae	1 (6.7%)
Putative place of acquisition	
Locally	12 (80%)
Abroad	3 (20%)

**Table 2 pathogens-13-00092-t002:** Description of symptoms and characteristics of the lesions.

Signs and Symptoms	N (%) or Median (IQR)
Symptoms at onset	
Fever	12 (80%)
Lymphadenopathy	9 (60%)
Pharyngitis	5 (33.3%)
Proctitis	10 (66.7%)
Skin/mucosal lesions	15 (100%)
Oral and perioral	3 (20%)
Genital	5 (33.3%)
Anal and perianal	6 (40%)
Number of lesions	
<5	2 (13.3%)
5–10	5 (33.3%)
>10	8 (53.3%)

**Table 3 pathogens-13-00092-t003:** Oropharyngeal and skin lesion swab results (and cycle threshold in brackets) among 11 patients followed longitudinally.

	Baseline		Week 2		Week 3		Week 4		Week 5	
	Pharynx (Ct)	Skin (Ct)	Pharynx (Ct)	Skin (Ct)	Pharynx (Ct)	Skin (Ct)	Pharynx (Ct)	Skin (Ct)	Pharynx (Ct)	Skin (Ct)
Patient 1	Pos (26)	Pos (19)	Pos (31)	Pos (32)	Pos (32)	Neg	-	-	Pos (26)	-
Patient 2	Pos (28)	Pos (22)	-	Neg	Pos (34)	-	Pos (32)	-	Neg	-
Patient 3	Pos (16)	Pos (13)	Pos (35)	Neg	Pos (33)	-	Neg	-	-	-
Patient 4	Pos (22)	Pos (17)	Neg	Pos (32)	-	Neg	-	-	-	-
Patient 5	Pos (19)	Pos (24)	Pos (21)	Pos (35)	Neg	Neg	-	-	-	-
Patient 6	Neg	Pos (34)	-	Neg	-	-	-	-	-	-
Patient 7	Pos (23)	Pos (16)	Neg	Neg						
Patient 8	Pos (25)	Pos (16)	Neg	Neg						
Patient 9	Pos (32)	Neg	Pos (35)	-	Neg	-				
Patient 10	Pos (26)	Pos (31)	Neg	Neg						
Patient 11	Pos (32)	Pos (16)	Neg	Neg						

## Data Availability

Data are available upon reasonable request to the corresponding author.

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
