# Peer review of "Mpox Virus in the Pharynx of Men Having Sex with Men: A Case Series"

_pathogens, 2024, doi:10.3390/pathogens13010092_

Round 1

Reviewer 1 Report

Comments and Suggestions for Authors

1. Line 30: 26.229 cases should be the error of 26,229.

2. Readers maybe cannot understand “Unspecified sex with other men”, perhaps, should the authors write "Patients refuse to reveal"?

3. Table 3 shows 11 patients, are the other 4 lost to follow-up? This should be introduced.

4. According to the description in lines 83-85, if there is one case detected solely by oropharyngeal swabs, then there should be two cases detected solely by skin lesions. Is this the case?

Author Response

We thank the reviewer for the invaluable insights provided by his revision.
Please find below a point-by-point response to his/her comments (in italic blue) 

Line 30: 26.229 cases should be the error of 26,229.

Line 30 has been duly corrected.

Readers maybe cannot understand “Unspecified sex with other men”, perhaps, should the authors write "Patients refuse to reveal"?

We thank the Reviewer for the comment.  It is important to note that all patients in our study were men who reported engaging in sexual activities with other men. While the specifics regarding the type of intercourse (whether receptive anal, receptive oral, or both) were elucidated for five individuals, this detail was unavailable for nine cases. In response to this concern, we revised the table, now designating the category as "Sex with other men (unspecified)" to enhance clarity.

Table 3 shows 11 patients, are the other 4 lost to follow-up? This should be introduced.

As mentioned in the materials, the longitudinal follow-up was performed only for patients included in the study after July 2022. Consequently, longitudinal data were available for only 11 out of 15 patients. This is now better elucidated in the revised manuscript (lines 94-97).

According to the description in lines 83-85, if there is one case detected solely by oropharyngeal swabs, then there should be two cases detected solely by skin lesions. Is this the case?

The reviewer's observation is correct. In two instances, the baseline oropharyngeal swab produced negative results. This information has been now explicitly stated in the manuscript (lines 86-87).

Reviewer 2 Report

Comments and Suggestions for Authors

This study presented 15 cases of acute MPXV infection and ct values of PCR results. The objective of this study is to show the dynamic of viral load of MPVX. However, presenting Ct value will be misleading to the readers. Ct values can be affected by many factors, including sample volume, cell numbers, viral load etc. The ct values should not be used for comparison between different samples sources.

Author Response

We thank the reviewer for raising such a pertinent concern, which we fully acknowledge. However, despite these acknowledged limitations, we would like to emphasize our decision to include Ct values in our study. We intend to provide an approximate estimation of Mpox DNA values across different biological samples. While we acknowledge the factors that can affect Ct values, we believe that presenting this data contributes to a broader understanding of viral load dynamics in different contexts. We, of course, ensured that these limitations were transparently communicated in the manuscript (see lines 136-153 in the revised version of the discussion)

Reviewer 3 Report

Comments and Suggestions for Authors

Limonta et al. report on a small number of individuals with pharyngeal manifestations of Mpox virus infections. I have a few suggestions on how the manuscripts may be further improved.

1.) I appreciated that the manuscript is short and straightforward, however, considering the preliminary nature of the research and the low number of study participants, a “correspondence” format seems appropriate. I’ll leave this to the editor’s discretion.

2.) Abstract) The authors suggest the assessment of the impact of human pharyngeal virus shedding on disease transmission. Presently, I do not have enough phantasy to imagine how such an assessment could be conducted in an ethically acceptable way. Accordingly, please either be more specific on this or remove the potentially misunderstandable point.

3.) Methods, lines 48-50: In line with scientific reporting standards, please mention town and country of the manufacturer of the applied PCR kit.

4.) Methods, lines 51-52: If first appearance of negative results led to cessation of the testing approach, potential relapse will necessary have gone undetected. The authors might at least want to mention this limitation in the discussion.

5.) Results: Some of the recorded symptoms like, e.g., lymphadenopathy (line 69) can be associated with the patients’ underlying diseases like, e.g., HIV infections as well. Th value of the text for the readers would be increased if the authors could outline such ambiguous situations. For example, was lymphadenopathy associated with pre-existing HIV infections? Or did it newly appeared or was it at least aggravated (in a measurable way) associated with the newly acquired mpox infection?

Author Response

Limonta et al. report on a small number of individuals with pharyngeal manifestations of Mpox virus infections. I have a few suggestions on how the manuscripts may be further improved.

We thank the Reviewer for the useful suggestions. Please find below a point-by-point answer, following the original comments (in italic blue)

1.) I appreciated that the manuscript is short and straightforward, however, considering the preliminary nature of the research and the low number of study participants, a “correspondence” format seems appropriate. I’ll leave this to the editor’s discretion.

We will defer the decision to transform our manuscript into correspondence to the discretion of the editor. We appreciate the reviewer's suggestion and trust that the editorial team will make the most informed and appropriate decision regarding the format of our manuscript.

2.) Abstract) The authors suggest the assessment of the impact of human pharyngeal virus shedding on disease transmission. Presently, I do not have enough phantasy to imagine how such an assessment could be conducted in an ethically acceptable way. Accordingly, please either be more specific on this or remove the potentially misunderstandable point.

We have removed the point as suggested by the Reviewer

3.) Methods, lines 48-50: In line with scientific reporting standards, please mention town and country of the manufacturer of the applied PCR kit.

We thank again the Reviewer for his/her suggestion. The missing information have now been included (line 50)

4.) Methods, lines 51-52: If first appearance of negative results led to cessation of the testing approach, potential relapse will necessary have gone undetected. The authors might at least want to mention this limitation in the discussion.

The concern raised by the reviewer is valid and appropriate. We have now included it as a limitation in the discussion of our results (lines 156-158)

5.) Results: Some of the recorded symptoms like, e.g., lymphadenopathy (line 69) can be associated with the patients’ underlying diseases like, e.g., HIV infections as well. Th value of the text for the readers would be increased if the authors could outline such ambiguous situations. For example, was lymphadenopathy associated with pre-existing HIV infections? Or did it newly appeared or was it at least aggravated (in a measurable way) associated with the newly acquired mpox infection?

All the lymphadenopathies and systemic symptoms were new-onset and unlikely to be related with well-controlled HIV infections. This is now better explained in lines 69-71

Round 2

Reviewer 2 Report

Comments and Suggestions for Authors

The authors called for further research for Mpox transmission. The authors may also add in the Discussion that the DNA replication and dynamic change of Ct value of Mpox in patients warrant further investigation as the current knowledge on the MPXV DNA replication is still limited, most of which was based on vaccinia virus research. It would be better to cite some references to support the author's points, for example, the below reference.

Liu B, Panda D, Mendez-Rios JD, Ganesan S, Wyatt LS, Moss B. Identification of Poxvirus Genome Uncoating and DNA Replication Factors with Mutually Redundant Roles. J Virol. 2018 Mar 14;92(7):e02152-17. doi: 10.1128/JVI.02152-17. PMID: 29343579; PMCID: PMC5972866.

Author Response

We would like to express our gratitude to the reviewer for their valuable comment. We have taken into consideration the limitation regarding the evidence on Mpox replication and Ct dynamics, which we included in the discussion (please see lines 143-144 of the current version). Additionally, we have added a reference to support our choice of reporting Ct as a proxy of viral replication of Mpox (now ref 14). 

Best regards, 

Giuseppe Lapadula & Silvia Limonta